# Personalising intravenous to oral antibiotic switch decision making through fair interpretable machine learning

William J. Bolton [1,2,3,4] ✉, Richard Wilson[1,4,5], Mark Gilchrist[1,4,6], Pantelis Georgiou[1,4,7], Alison Holmes[1,4,5,8] & Timothy M. Rawson [1,4]

Antimicrobial resistance (AMR) and healthcare associated infections pose a significant threat globally. One key prevention strategy is to follow antimicrobial stewardship practices, in particular, to maximise targeted oral therapy and reduce the use of indwelling vascular devices for intravenous (IV) administration. Appreciating when an individual patient can switch from IV to oral antibiotic treatment is often non-trivial and not standardised. To tackle this problem we created a machine learning model to predict when a patient could switch based on routinely collected clinical parameters. 10,362 unique intensive care unit stays were extracted and two informative feature sets identified. Our best model achieved a mean AUROC of 0.80 (SD 0.01) on the hold-out set while not being biased to individuals protected characteristics. Interpretability methodologies were employed to create clinically useful visual explanations. In summary, our model provides individualised, fair, and interpretable predictions for when a patient could switch from IV-to-oral antibiotic treatment. Prospectively evaluation of safety and efficacy is needed before such technology can be applied clinically.

Antimicrobial stewardship aims to optimise drug use to prolong current therapeutic effectiveness and combat antimicrobial resistance (AMR)[1]. One key aspect of antimicrobial stewardship is the route of administration. It is common for critically ill patients to be given empirical intravenous (IV) antibiotic therapy upon admission due to rapid delivery, high bioavailability, and uncertainty surrounding a potential infection. Then later in the treatment regime once the patient is stabilized and the infection is understood, their antibiotics are often switched to an oral administration route. There is a well described focus to switch from IV-to-oral administration as early as possible and to use more oral drugs when appropriate, given they are often equally effective and can reduce side effects during prolonged exposure[2-5]. In a range of infectious diseases that were traditionally treated with IV only (e.g., bacteremia, endocarditis, and bone and joint infections), recent studies have demonstrated that oral therapy can be non-inferior to IV in efficacy[6-11]. Furthermore, reducing the unnecessary use of indwelling IV devices is a well established patient safety and infection prevention priority to minimise the risk of healthcare associated infections (HCAIs)[12]. Beyond the infection complications of IV catheters, oral administration is more comfortable for the patient, reduces nurses' workload, and allows for easy discharge from the hospital. Furthermore, oral therapy is cheaper and more cost-effective[13]. The UK Health Security Agency recently published national antimicrobial IV-to-oral switch (IVOS) criteria for early switching[14].

[1]Centre for Antimicrobial Optimisation, Imperial College London, London, UK. [2]AI4Health Centre for Doctoral Training, Imperial College London, London, UK. [3]Department of Computing, Imperial College London, London, UK. [4]National Institute for Health Research, Health Protection Research Unit in Healthcare Associated Infections and Antimicrobial Resistance, Imperial College London, London, UK. [5]Faculty of Health & Life Sciences, University of Liverpool, Liverpool, UK. [6]Imperial College Healthcare NHS Trust, London, UK. [7]Centre for Bio-inspired Technology, Department of Electrical and Electronic Engineering, Imperial College London, London, UK. [8]Department of Infectious Diseases, Imperial College London, London, UK. ✉e-mail: william.bolton@imperial.ac.uk

The requirements were developed based on expert consensus and primarily revolve around the patient's clinical and infection markers improving as well as specific points with regards to absorption, bioavailability, and infection type.

Despite significant evidence, the uptake of early oral therapy remains low[15,16], and beyond guidelines[14] there is a lack of research in IV-to-oral decision support systems. Given this, we decided to investigate if a machine learning based clinical decision support system (CDSS) could assist antibiotic switch decision making at the individual patient level. More specifically neural network models were developed to predict, based on routinely collected clinical parameters, whether a patient could be suitable for switching from IV-to-oral antibiotics on any given day. ICU data was utilised given it is widely available, comprehensive, and if a CDSS can be developed for critical patients then it can likely be adapted to less severe settings. Many CDSSs utilising machine learning have been developed to assist with other aspects of antimicrobial use[17–19]; however, limited clinical utilisation and adoption has been seen[20]. As such, when tackling this problem we wanted to ensure our CDSS solution was simple, fair, interpretable, and generalisable to maximise the ability for clinical translation. By simple we mean the model architecture can be understood by non-experts, while fair infers model performance is not biased to particular sensitive attributes or protected characteristics. Interpretability means predictions can more easily be understood, explained, and trusted. Finally, a model is generalisable if it can be applied to many healthcare settings with consistent performance. We imagine by providing individualised antibiotic switch estimations such a system could support patient-centric decisions and provide assurance on if switching could be appropriate or not in a given clinical context. Figure 1 shows an overview of this research.

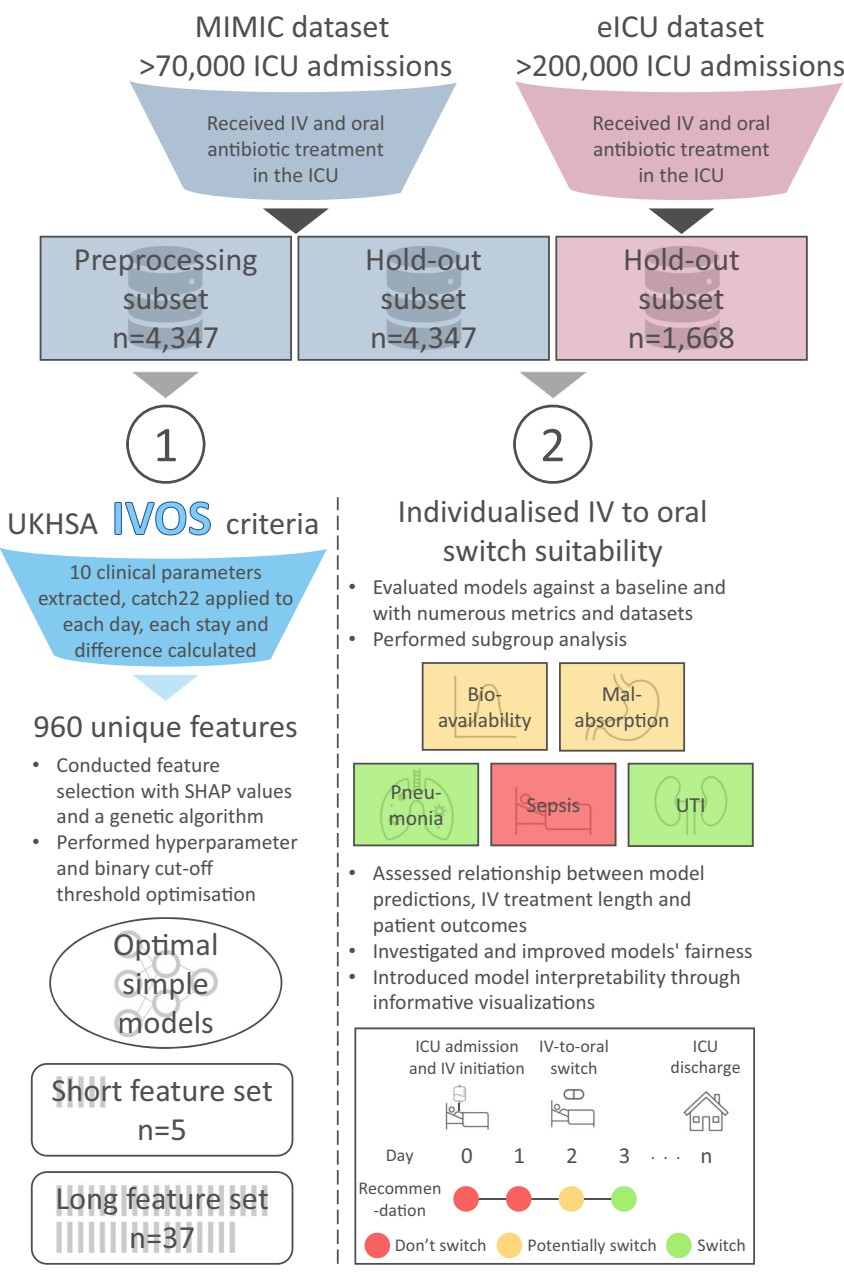

**Fig. 1 | Overview of the steps taken in this research study to develop fair and interpretable machine learning models for antimicrobial switch decision making.** MIMIC Medical Information Mart for Intensive Care, ICU Intensive care unit, IV Intravenous, IVOS Intravenous-to-oral switch, SHAP SHapley Additive exPlanations, UTI Urinary tract infection.

**Table 1 | Dataset demographics and statistics**

| Statistic | | Dataset | | |
|---|---|---|---|---|
| | | MIMIC preprocessing | MIMIC hold-out | eICU |
| Number of stays | | 4347 | 4347 | 1668 |
| Age (mean) | | 65.30 (SD 15.16) | 65.44 (SD 15.23) | 64.74 (SD 15.91) |
| Length of stay (mean) | | 3.14 (SD 2.78) | 3.12 (SD 2.71) | 3.17 (SD 2.83) |
| Sex (%) | Male | 58.82 | 58.92 | 50.30 |
| | Female | 41.18 | 41.08 | 49.70 |
| Race (%) | White | 67.97 | 68.09 | 78.77 |
| | Black | 9.55 | 9.92 | 15.08 |
| | Unknown | 10.28 | 9.35 | 3.50 |
| | Other | 6.22 | 5.48 | |
| | Hispanic | 3.27 | 3.94 | 1.39 |
| | Asian | 2.34 | 2.94 | 0.84 |
| | Native American | 0.36 | 0.27 | 0.42 |
| Antimicrobial treatment length (mean) | Overall | 3.34 (SD 2.16) | 3.29 (SD 2.01) | 2.97 (SD 1.94) |
| | IV | 2.79 (SD 1.47) | 2.75 (SD 1.46) | 2.46 (SD 1.55) |
| | Oral | 2.76 (SD 1.90) | 2.69 (SD 1.81) | 1.14 (SD 1.78) |
| Infection type (%, most common shown) | UTI | 65.10 | 64.69 | 10.19 |
| | Pneumonia | 26.30 | 26.62 | 31.71 |
| | Sepsis | 18.20 | 19.63 | 32.91 |
| IV antibiotics (%, those with a frequency of greater than 5% shown) | Vancomycin | 35.96 | 35.56 | 41.98 |
| | Cefepime | 12.28 | 14.03 | 2.86 |
| | Cefazolin | 12.47 | 12.08 | 1.71 |
| | Piperacillin-Tazobactam | 9.35 | 8.42 | 8.92 |
| | Ceftriaxone | 7.82 | 8.04 | 6.10 |
| | Levofloxacin | 1.76 | 1.99 | 14.12 |
| Oral antibiotics (%, those with a frequency of greater than 5% shown) | Azithromycin | 22.96 | 23.18 | 12.24 |
| | Vancomycin | 13.45 | 14.23 | 12.37 |
| | Ciprofloxacin | 12.77 | 12.14 | - |
| | Levofloxacin | 12.69 | 12.37 | 51.70 |
| | Sulfameth/Trimethoprim | 11.82 | 10.04 | - |
| | Metronidazole | 7.90 | 9.32 | 22.72 |
| Microbiology (%, those with a frequency of greater than 5% shown) | Positive growth | 32.76 | 33.07 | 13.01 |
| | *Escherichia coli* | 6.86 | 6.51 | - |
| | *Staphylococcus aureus* | 6.69 | 7.33 | - |

Note that infection types in MIMIC are determined through 'hadm_id' and are not definitively linked to the antimicrobial switch of interest as diagnoses are only coded for billing purposes upon hospital discharge. This results in totals over 100% as patients have multiple infection episodes. Sex was determined based on the information contained within the dataset. *MIMIC* Medical Information Mart for Intensive Care, *ICU* Intensive care unit, *IV* Intravenous, *SD* Standard deviation, *UTI* Urinary tract infection.

## Results

### Data

8694 unique intensive care unit (ICU) stays, were extracted from the MIMIC dataset[21,22], with 1,668 from eICU[22–24]. 10 clinical features were selected based on the UK antimicrobial IVOS criteria[14] (Supplementary Table. 1). Transformation of those temporally dynamic values into time series features as detailed in Section 4.1 resulted in 960 unique features for each day of each patient's stay. Details of the MIMIC preprocessing, MIMIC hold-out, and eICU datasets are shown in Table 1. All the datasets are relatively equally balanced, however, the specific antibiotic utilisation distribution varies between MIMIC and eICU. Furthermore, eICU represents a more unwell population given the higher proportion of life-threatening infections such as sepsis.

The antibiotic spectrum index (ASI)[25] demonstrates if an antibiotic treatment regime shows broad or narrow activity. Larger values indicate a broader spectrum, while smaller values correlate with more targeted activity. A statistically significant ($p$-value < 0.01, statistic 1686390, alpha 0.05, effect size 0.87) difference was found between the mean ASI for IV and oral antibiotics upon switching (8.25 and 5.89 respectively) through the Wilcoxon rank-sum test. In addition, the majority of patients (70.03%) see a decrease in their treatments ASI upon switching, with a mean decrease of 23.04% although this was highly variable (Supplementary Fig. 1).

### Feature and model optimisation

The first excessively large neural network trained on the preprocessing training subset achieved an Area Under the Receiver Operating Characteristic curve (AUROC) of 0.76 on the preprocessing test subset. SHapley Additive exPlanations (SHAP) values[26] were calculated and the top 98 features were selected for input into a genetic algorithm. The genetic algorithm produced two sets of features, one short set, containing only 5 features, and another longer set of 37. The short and long feature sets achieved an AUROC of 0.80 and 0.82 on the preprocessing test subset respectively. The final features for each set are shown in

Supplementary Table. 1 with the respective Catch22 time-series transformations listed in Supplementary Table. 2. During hyperparameter optimisation our objective was to find the most simple models whilst maintaining performance. For both feature sets, this was achieved with new less complex models being found to achieve the same AUROC. The final hyperparameters of each model are shown in Supplementary Table. 3. Finally, alternative cutoff thresholds were explored for both models to maximise the AUROC and minimise the FPR (Supplementary Fig. 2). This then allows for a traffic light system to be employed at deployment for simplicity and interpretability (Fig. 4). Youden's index[27] which optimises the AUROC, found the 1st cutoff point of 0.54 and 0.52 for the short and long models respectively. The point where precision, recall and the F1 score were equal acted as a 2nd stringent threshold. This cutoff point was 0.74 for the short models and 0.79 for the long models, resulting in a lower AUROC (0.70 and 0.74 respectively) on the preprocessing test subset but a superior false positive rate (FPR) (0.11 and 0.09 respectively versus 0.26 and 0.22 using Youden's threshold).

## Model evaluation

The final short models trained and tested on the hold-out set obtained a mean AUROC of 0.78 (SD 0.02), FPR 0.25 (SD 0.02) with the 1st Youden's threshold, and a mean AUROC of 0.69 (SD 0.03), FPR 0.10 (SD 0.02) with the 2nd threshold. Meanwhile, the final long models achieved a mean AUROC of 0.80 (SD 0.01), FPR 0.25 (SD 0.04) with the 1st cutoff, and a mean AUROC of 0.75 (SD 0.02), FPR 0.10 (SD 0.03) with the 2nd cutoff. Further evaluation metrics for each model and threshold can be found in Table 2. For comparison, a baseline that utilised two clear infection markers (temperature and Early Warning Score) from the latest guidelines[14] obtained worse results with an AUROC of 0.66, accuracy of 0.61, TPR of 0.75, and FPR of 0.43. Predictions and labels broken down by IV treatment duration (Fig. 2) shows that the majority of incorrect predictions occurred in the middle of IV treatment days when the models predicted to switch but the real label indicated the patient continued with IV. The short model on average predicted 70% and 38% of patients could switch earlier than they did with the 1st and 2nd thresholds respectively. Arguably the long model demonstrated a more balanced profile with 51/28% early, 38/41% agreement, and 11/31% late switch predictions with the 1st and 2nd thresholds. When the difference between the real and predicted switch event was minimal, mean patient LOS outcomes were reduced (Fig. 3). Furthermore, a statistically significant difference (Wilcoxon rank-sum test, alpha 0.05) in remaining LOS was observed between those who received oral versus those who had IV treatment, with 2, 3, and 4 prior days of IV treatment (oral mean, IV mean, *p*-value, statistic and effect size of 1.03;1.70; < 0.01;555588;0.39, 0.91;1.89; < 0.01;227473;0.56 and 0.95;2.02; < 0.01;24572;0.57 respectively). No statistically significant differences were observed on the later days 5, 6, and 7 (Supplementary Fig. 3). No mortality differences were observed due to imbalanced data (Supplementary Table. 4).

eICU is a different dataset from MIMIC covering distinct hospitals with a separate patient population and unique data distribution. These differences can often cause problems for machine learning models but allows us to validate our features and modeling approach on an external dataset. When applied to eICU data via transfer learning a mean AUROC of 0.72 (SD 0.02), 0.65 (SD 0.05), 0.72 (SD 0.02), 0.64 (SD 0.06), and a FPR of 0.24 (SD 0.04), 0.05 (SD 0.02), 0.24 (SD 0.04) and 0.06 (SD 0.03) was obtained for the short and long models 1st and 2nd thresholds respectively (Table 2). Both models outperformed the eICU baseline which obtained an AUROC of 0.55, accuracy of 0.67, TPR of 0.38, and FPR 0.28.

Achieving target drug exposure against the pathogenic organism is important during antibiotic treatment and is often a concern when deciding to switch to oral administration[28]. For those patients who were on oral antibiotics with incomplete absorption a mean AUROC of 0.73 (SD 0.03), 0.67 (SD 0.05), 0.77 (SD 0.02), 0.73 (SD 0.03), and a FPR of 0.33 (SD 0.06), 0.12 (SD 0.04), 0.28 (SD 0.07) and 0.12 (SD 0.07) was achieved for the short and long models 1st and 2nd cutoffs respectively (Table 2).

If patients have issues with enteral absorption, oral antibiotic therapy is less likely to be suitable[14]. When tested on patients with poor absorption a mean AUROC of 0.76 (SD 0.10), 0.75 (SD 0.11), 0.75 (SD 0.07), 0.71 (SD 0.16), and a FPR of 0.48 (SD 0.20), 0.28 (SD 0.12), 0.43 (SD 0.14) and 0.12 (SD 0.12) was obtained for the short and long models 1st and 2nd thresholds respectively (Table 2).

Results were then examined for patients with specific infections. For urinary tract infection (UTI) patients a mean AUROC of 0.77 (SD 0.03), 0.74 (SD 0.04), 0.78 (SD 0.02), 0.77 (SD 0.04), and an FPR of 0.33 (SD 0.03), 0.15 (SD 0.03), 0.31 (SD 0.04) and 0.13 (SD 0.05) was achieved for the short and long models 1st and 2nd cutoffs respectively (Table 2). When tested on patients with pneumonia a mean AUROC of 0.76 (SD 0.03), 0.76 (SD 0.03), 0.77 (SD 0.02), 0.74 (SD 0.04), and a FPR of 0.35 (SD 0.03), 0.16 (SD 0.04), 0.32 (SD 0.04) and 0.14 (SD 0.04) was obtained for the short and long models 1st and 2nd thresholds respectively (Table 2). Finally, for sepsis patients, a mean AUROC of 0.82 (SD 0.05), 0.79 (SD 0.12), 0.77 (SD 0.07), 0.76 (SD 0.18), and a FPR of 0.36 (SD 0.10), 0.17 (SD 0.09), 0.35 (SD 0.08) and 0.16 (SD 0.07) was achieved for the short and long models 1st and 2nd cutoffs respectively (Table 2).

## Interpretability

Two cutoff thresholds allows for a simple traffic light system to be presented to clinicians with regards to if a switch could be appropriate at a particular time. To further improve interpretability and model understanding the SimplEx[29] methodology was applied. Once fitted the decomposition for a particular patient was computed to get corpus examples, their importance, and feature contribution. This data was combined and infectious disease clinicians consulted to create informative visual representations. Figure 4 shows an example of these for short model predictions.

## Fairness

Overall the models demonstrated equalised odds (EO) across the majority of sensitive attribute groups. Table 3 shows the AUROC, TPR, and FPR for both short and long models by sensitive attribute group. The short model did not obtain EO for those in the age bracket of 90, of Native American descendance, or with Medicaid insurance (Table 3). On the other hand, the long model only showed a discrepancy for patients in the age bracket of 30. For the short model, threshold optimisation[30] with the true positive rate (TPR) parity constraint enabled EO to be achieved for those in the age bracket of 90, while the EO constraint standardised performance across insurance groups (Supplementary Table. 5, Supplementary Fig. 4). No constraint enabled the model to demonstrate EO for the native group. For the long model, the FPR parity constraint caused EO to be obtained for those in the age bracket of 30 (Supplementary Table. 6, Supplementary Fig. 4).

## Discussion

To maximise clinical utility we aimed to minimise complexity during feature selection and model development. Through the genetic algorithm, two feature sets of interest were identified. The short set utilised only 5 features but maintained performance, while the long set enabled slight improvements in the evaluation metrics. The two most important SHAP features utilised the same time series transformation (SB_MotifThree_quantile_hh) for systolic blood pressure over the whole ICU stay and heart rate over the current day respectively. This measure uses equiprobable binning to indicate the predictability of a time series. This is medically relevant to switching the administration route as clinicians look for vitals to stabilise before switching. Interestingly the 3rd and 4th SHAP ranked features represent the same type

**Table 2 | Evaluation results**

| Dataset | Model | Threshold | Evaluation Metric | | | | | | |
|---|---|---|---|---|---|---|---|---|---|
| | | | AUROC | Accuracy | Precision | F1 Score | AUPRC | TPR | FPR |
| MIMIC | Short | 1st | 0.78 (SD 0.02) | 0.76 (SD 0.01) | 0.39 (SD 0.02) | 0.53 (SD 0.03) | 0.35 (SD 0.03) | 0.80 (SD 0.05) | 0.25 (SD 0.02) |
| | | 2nd | 0.69 (SD 0.03) | 0.83 (SD 0.01) | 0.49 (SD 0.03) | 0.48 (SD 0.04) | 0.32 (SD 0.03) | 0.48 (SD 0.06) | 0.10 (SD 0.02) |
| | Long | 1st | 0.80 (SD 0.01) | 0.77 (SD 0.03) | 0.41 (SD 0.04) | 0.55 (SD 0.03) | 0.37 (SD 0.03) | 0.85 (SD 0.04) | 0.25 (SD 0.04) |
| | | 2nd | 0.75 (SD 0.02) | 0.85 (SD 0.02) | 0.55 (SD 0.05) | 0.57 (SD 0.03) | 0.40 (SD 0.03) | 0.61 (SD 0.06) | 0.10 (SD 0.03) |
| MIMIC Incomplete absorption | Short | 1st | 0.73 (SD 0.03) | 0.72 (SD 0.03) | 0.64 (SD 0.04) | 0.71 (SD 0.03) | 0.59 (SD 0.03) | 0.79 (SD 0.06) | 0.33 (SD 0.06) |
| | | 2nd | 0.67 (SD 0.05) | 0.71 (SD 0.04) | 0.74 (SD 0.06) | 0.56 (SD 0.11) | 0.57 (SD 0.05) | 0.47 (SD 0.12) | 0.12 (SD 0.04) |
| | Long | 1st | 0.77 (SD 0.02) | 0.76 (SD 0.03) | 0.68 (SD 0.05) | 0.74 (SD 0.03) | 0.63 (SD 0.04) | 0.82 (SD 0.06) | 0.28 (SD 0.07) |
| | | 2nd | 0.73 (SD 0.03) | 0.75 (SD 0.03) | 0.79 (SD 0.09) | 0.66 (SD 0.06) | 0.63 (SD 0.05) | 0.57 (SD 0.09) | 0.12 (SD 0.07) |
| MIMIC Mal- absorption | Short | 1st | 0.76 (SD 0.10) | 0.64 (SD 0.17) | 0.44 (SD 0.25) | 0.57 (SD 0.23) | 0.44 (SD 0.25) | 1.00 (SD 0.00) | 0.48 (SD 0.21) |
| | | 2nd | 0.75 (SD 0.11) | 0.73 (SD 0.06) | 0.41 (SD 0.20) | 0.50 (SD 0.21) | 0.39 (SD 0.15) | 0.78 (SD 0.33) | 0.28 (SD 0.12) |
| | Long | 1st | 0.75 (SD 0.07) | 0.65 (SD 0.10) | 0.39 (SD 0.16) | 0.53 (SD 0.17) | 0.38 (SD 0.15) | 0.93 (SD 0.13) | 0.43 (SD 0.14) |
| | | 2nd | 0.71 (SD 0.16) | 0.79 (SD 0.10) | 0.57 (SD 0.35) | 0.48 (SD 0.26) | 0.45 (SD 0.24) | 0.53 (SD 0.34) | 0.12 (SD 0.12) |
| MIMIC UTI | Short | 1st | 0.77 (SD 0.03) | 0.71 (SD 0.03) | 0.38 (SD 0.07) | 0.52 (SD 0.07) | 0.35 (SD 0.07) | 0.87 (SD 0.05) | 0.33 (SD 0.05) |
| | | 2nd | 0.74 (SD 0.04) | 0.81 (SD 0.02) | 0.49 (SD 0.08) | 0.54 (SD 0.08) | 0.38 (SD 0.08) | 0.63 (SD 0.10) | 0.15 (SD 0.03) |
| | Long | 1st | 0.78 (SD 0.02) | 0.72 (SD 0.03) | 0.39 (SD 0.06) | 0.54 (SD 0.06) | 0.36 (SD 0.06) | 0.87 (SD 0.04) | 0.31 (SD 0.04) |
| | | 2nd | 0.77 (SD 0.04) | 0.83 (SD 0.02) | 0.54 (SD 0.09) | 0.58 (SD 0.07) | 0.42 (SD 0.07) | 0.66 (SD 0.12) | 0.13 (SD 0.05) |
| MIMIC Pneumonia | Short | 1st | 0.76 (SD 0.03) | 0.68 (SD 0.03) | 0.28 (SD 0.05) | 0.42 (SD 0.07) | 0.26 (SD 0.05) | 0.86 (SD 0.07) | 0.35 (SD 0.03) |
| | | 2nd | 0.76 (SD 0.03) | 0.82 (SD 0.02) | 0.41 (SD 0.05) | 0.51 (SD 0.04) | 0.32 (SD 0.05) | 0.68 (SD 0.07) | 0.16 (SD 0.03) |
| | Long | 1st | 0.77 (SD 0.02) | 0.70 (SD 0.04) | 0.30 (SD 0.06) | 0.44 (SD 0.07) | 0.28 (SD 0.05) | 0.87 (SD 0.03) | 0.32 (SD 0.04) |
| | | 2nd | 0.74 (SD 0.04) | 0.82 (SD 0.03) | 0.41 (SD 0.09) | 0.48 (SD 0.08) | 0.31 (SD 0.07) | 0.62 (SD 0.10) | 0.14 (SD 0.04) |
| MIMIC Sepsis | Short | 1st | 0.82 (SD 0.05) | 0.71 (SD 0.05) | 0.38 (SD 0.15) | 0.53 (SD 0.17) | 0.38 (SD 0.15) | 1.00 (SD 0.00) | 0.36 (SD 0.10) |
| | | 2nd | 0.79 (SD 0.12) | 0.82 (SD 0.08) | 0.53 (SD 0.25) | 0.56 (SD 0.20) | 0.43 (SD 0.18) | 0.76 (SD 0.27) | 0.17 (SD 0.09) |
| | Long | 1st | 0.77 (SD 0.07) | 0.68 (SD 0.08) | 0.35 (SD 0.14) | 0.48 (SD 0.14) | 0.33 (SD 0.13) | 0.90 (SD 0.13) | 0.35 (SD 0.08) |
| | | 2nd | 0.76 (SD 0.18) | 0.80 (SD 0.10) | 0.46 (SD 0.21) | 0.52 (SD 0.22) | 0.42 (SD 0.16) | 0.68 (SD 0.33) | 0.16 (SD 0.07) |
| eICU | Short | 1st | 0.72 (SD 0.02) | 0.75 (SD 0.03) | 0.36 (SD 0.05) | 0.47 (SD 0.05) | 0.29 (SD 0.04) | 0.68 (SD 0.06) | 0.24 (SD 0.04) |
| | | 2nd | 0.65 (SD 0.05) | 0.85 (SD 0.02) | 0.60 (SD 0.12) | 0.42 (SD 0.10) | 0.31 (SD 0.06) | 0.34 (SD 0.10) | 0.05 (SD 0.02) |
| | Long | 1st | 0.72 (SD 0.02) | 0.75 (SD 0.02) | 0.36 (SD 0.06) | 0.46 (SD 0.04) | 0.29 (SD 0.04) | 0.67 (SD 0.07) | 0.24 (SD 0.04) |
| | | 2nd | 0.64 (SD 0.06) | 0.84 (SD 0.02) | 0.48 (SD 0.17) | 0.38 (SD 0.15) | 0.28 (SD 0.07) | 0.33 (SD 0.15) | 0.06 (SD 0.03) |

MIMIC;Medical Information Mart for Intensive Care, ICU;Intensive care unit, UTI;Urinary tract infection, SD;Standard deviation, AUROC;Area under the receiver operating characteristic, AUPRC;Area under precision-recall, TPR;True positive rate, FPR;False positive rate

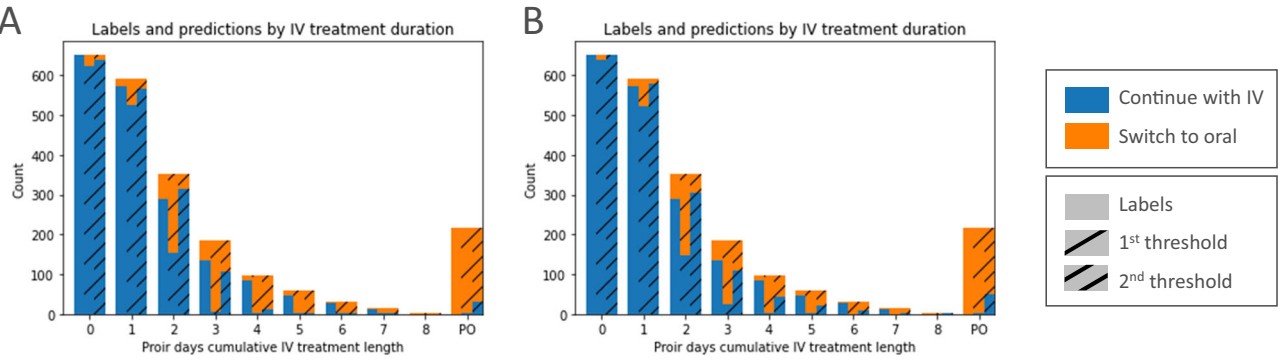

**Fig. 2 | Labels and predictions by IV treatment duration.** Plots for the short model **A** and the long model **B**. IV Intravenous.

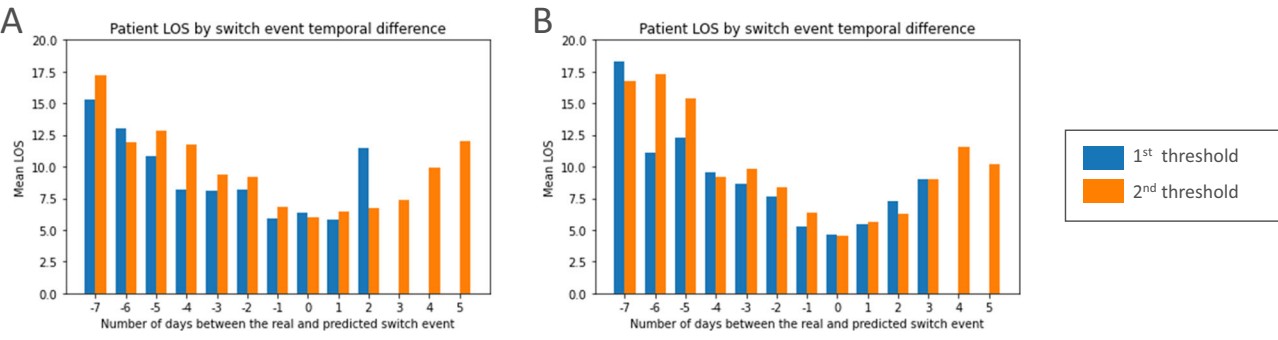

**Fig. 3 | Mean patient LOS outcomes by days between the real and predicted switch event.** Plots for the short model **A** and the long model **B**. A negative number on the *x* axis indicates the predicted switch event was before the real switch event. The opposite is true for positive numbers, while 0 means they occurred on the same day. IV Intravenous, LOS Length of stay.

of feature (IN_AutoMutualInfoStats_40_gaussian_fmmi calculated over the whole of the ICU stay) for two different clinical parameters, respiratory rate, and the mean blood pressure. Furthermore, their feature values (shown in Supplementary Fig. 5) are very similar, indicating why having both features was likely redundant to the models. Other features in the short set also demonstrate clinical importance. For example, the first minimum of an O2 saturation pulseoximetry autocorrelation function (CO_FirstMin_ac) would indicate variable stability and hence clinical improvement or deterioration. While a value indicating the importance of low frequencies in the Glasgow Coma Scale (GCS) motor response (SP_Summaries_welch_rect_area_5_1), could show if the patient is retaining consciousness or not over a long period, which is often necessary for administering oral medication. Overall these features combine to provide a comprehensive but succinct overview of the general health status of the patient which can be used to determine if switching could be appropriate.

Results on specific infections and antibiotic characteristics demonstrate the models have stable performance across numerous different patient groups. Particularly important is understanding when oral antibiotics with incomplete absorption can be used, given concerns surrounding achieving therapeutic concentrations. Our long model achieved an AUROC of 0.77 (SD 0.02) in this subpopulation. Furthermore, in conditions such as sepsis where patients are critically-ill for prolonged periods and fewer oral therapies are utilised, our short model obtained an AUROC of 0.82 (SD 0.02). Indicating that such a support system could be utilised in severe infections. Transfer learning results on the eICU dataset were stringent with regards to predicting when switching could be appropriate (Supplementary Fig. 3), this is to be expected considering the patients in eICU are on average more severely unwell than in MIMIC (Table 1). Switching administration route is influenced by many behavioural factors that are not easily modeled. Given eICU contains data from many different hospitals the

prescribing behaviour with regards to oral switching is likely much more heterogeneous than in MIMIC whose data is from a single institution. As such, the eICU model is having to approximate many different behaviours, which results in varying performance across institutions (Supplementary Fig. 6), and likely causes it to be more stringent with regard to predicting a switch to optimise performance. Similar behavior is observed with the baseline eICU results which confirms predicting the route of administration is a more challenging task in eICU when compared to MIMIC (AUROC of 0.66 and 0.55 respectively). Further research into subpopulations and other datasets could identify unfavourable IV-to-oral switch characteristics, such as individuals with abnormal pharmacokinetics or immunosuppression. Specific thresholding or separate models[31] could then ensure patients with such attributes require a larger output to be flagged as suitable for switching. Combining this with alternative thresholds to ensure fairness though can very quickly make CDSSs excessively complex, leading to misunderstanding, misuse, and reluctant adoption[17,20,32]. We believe this research strikes a practical balance between performance and usefulness for IV-to-oral switch decision support. Overall the results demonstrate our methods and models are generalisable as similar performance was obtained across all MIMIC tests with different patient populations, and between two distinct ICU datasets indicating the feature sets identified are informative and that the selected hyperparameters can model the underlying data.

Overall the models demonstrated reasonably fair performance across all sensitive attribute groups. When equalised odds were not achieved, threshold optimisation[30] was able to improve the results for a given group in all cases, except for that of the Native American group. This population was the most underrepresented within the data with an average of only 11 patients in the test set, highlighting the need for further good quality real or synthetic data on minority populations. When threshold optimisation was undertaken a trade-off between groups in a sensitive attribute class was sometimes observed. For

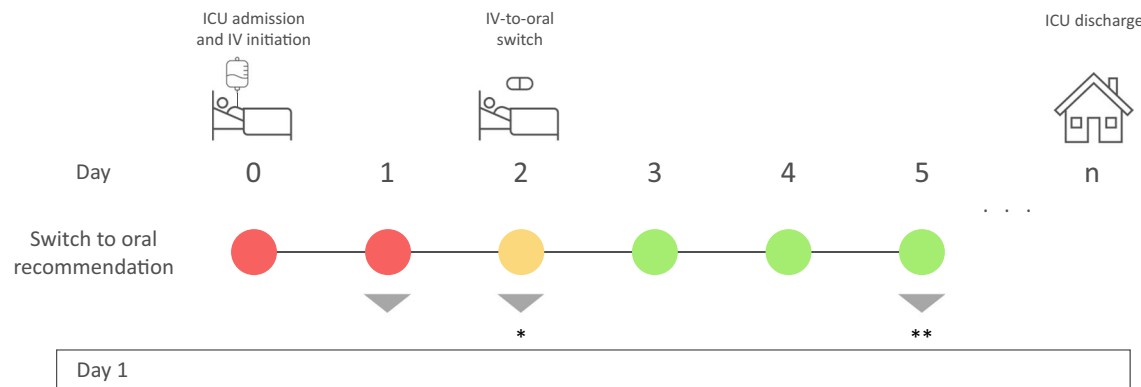

**Day 1**

Highlights
- Both thresholds predict switching is likely **not appropriate** at this time
- Predictions were correct for **100%** of similar examples
- O2 saturation pulseoximetry (feature 4) was of particular interest for these predictions

| | | Feature | | | | | Switch to oral label | Switch to oral prediction | |
| --- | --- | --- | --- | --- | --- | --- | --- | --- | --- |
| | Importance | 1 | 2 | 3 | 4 | 5 | | 1st threshold | 2nd threshold |
| Patient | - | 0.32 | 0.51 | 0.37 | 0.50 | 0.41 | 0 | 0 | 0 |
| Example 1 | 0.28 | 0.38 | 0.54 | 0.29 | 0.48 | 0.46 | 0 | 0 | 0 |
| 2 | 0.25 | 0.31 | 0.55 | 0.28 | 0.51 | 0.50 | 0 | 0 | 0 |
| 3 | 0.21 | 0.29 | 0.52 | 0.45 | 0.52 | 0.46 | 0 | 0 | 0 |
| 4 | 0.13 | 0.32 | 0.55 | 0.36 | 0.51 | 0.00 | 0 | 0 | 0 |

**Day 2**

\*

Highlights
- **Clinical guidance should be sought**, model thresholds disagree on whether switching could be appropriate or not at this time
- Predictions were correct for **50%** of similar examples (0% for the 1st threshold and 100% for the 2nd threshold)
- O2 saturation pulseoximetry (feature 4) was of particular interest for these predictions

| | | Feature | | | | | Switch to oral label | Switch to oral prediction | |
| --- | --- | --- | --- | --- | --- | --- | --- | --- | --- |
| | Importance | 1 | 2 | 3 | 4 | 5 | | 1st threshold | 2nd threshold |
| Patient | - | 0.24 | 0.25 | 0.28 | 0.43 | 0.77 | 1 | 1 | 0 |
| Example 1 | 0.38 | 0.25 | 0.20 | 0.25 | 0.42 | 0.73 | 0 | 1 | 0 |
| 2 | 0.12 | 0.21 | 0.12 | 0.20 | 0.43 | 0.85 | 0 | 1 | 0 |

**\*\* Day 5**

Highlights
- Both thresholds predict switching could be **appropriate** at this time
- Predictions were correct for **75%** of similar examples (75% for the 1st threshold and 75% for the 2nd threshold)
- Systolic blood pressure (feature 1) and O2 saturation pulseoximetry (feature 4) were of particular interest for these predictions

| | | Feature | | | | | Switch to oral label | Switch to oral prediction | |
| --- | --- | --- | --- | --- | --- | --- | --- | --- | --- |
| | Importance | 1 | 2 | 3 | 4 | 5 | | 1st threshold | 2nd threshold |
| Patient | - | 0.16 | 0.49 | 0.45 | 0.37 | 0.59 | 1 | 1 | 1 |
| Example 1 | 0.21 | 0.20 | 0.58 | 0.39 | 0.37 | 0.45 | 1 | 1 | 1 |
| 2 | 0.20 | 0.15 | 0.47 | 0.43 | 0.36 | 0.70 | 1 | 1 | 1 |
| 3 | 0.16 | 0.16 | 0.43 | 0.48 | 0.36 | 0.76 | 1 | 1 | 1 |
| 4 | 0.15 | 0.18 | 0.49 | 0.42 | 0.38 | 0.59 | 0 | 1 | 1 |

Note this system does not cover all aspects of the switch decision making process and should only be used as decision support to highlight when a patient may be suitable for switch assessment

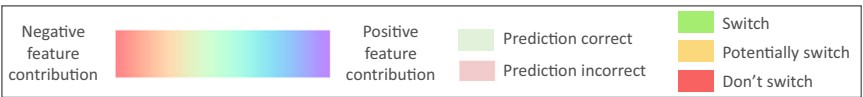

**Fig. 4 | Example visual representation for a particular patient.** `Traffic light' suggestions are initially displayed in a temporal manner as the patient progresses. If required clinicians can obtain more information for any given day of the patient's stay. Simple textual descriptions are provided alongside detailed tabular graphics to maximise clarity. The text quickly gets across key points, while tables show similar example patients and their features, both with their relative importance and contribution respectively. Furthermore, switch labels and predictions across both thresholds are displayed. Note that the patients label would obviously not be available during clinical use, but is shown here to be comprehensive. Finally, the limitations and use cases for the system are clearly labeled. ICU Intensive care unit, IV Intravenous.

**Table 3 | Fairness results**

| Sensitive attribute | Group | Short model | | | | Long model | | | |
|---|---|---|---|---|---|---|---|---|---|
| | | AUROC | TPR | FPR | EO | AUROC | TPR | FPR | EO |
| — | All patients | 0.78 (SD 0.02) | 0.80 (SD 0.05) | 0.25 (SD 0.02) | — | 0.80 SD (0.01) | 0.85 SD (0.04) | 0.25 SD (0.04) | — |
| Sex | Female | 0.74 (SD 0.18) | 0.79 (SD 0.35) | 0.29 (SD 0.13) | ✓ | 0.80 SD (0.15) | 0.85 SD (0.34) | 0.25 SD (0.10) | ✓ |
| | Male | 0.80 (SD 0.08) | 0.82 (SD 0.22) | 0.23 (SD 0.06) | ✓ | 0.80 SD (0.16) | 0.84 SD (0.33) | 0.23 SD (0.08) | ✓ |
| Age | 20 | 0.73 (SD 0.08) | 0.74 (SD 0.15) | 0.27 (SD 0.06) | ✓ | 0.76 SD (0.09) | 0.77 SD (0.16) | 0.24 SD (0.05) | ✓ |
| | 30 | 0.80 (SD 0.02) | 0.86 (SD 0.06) | 0.26 (SD 0.04) | ✓ | 0.72 SD (0.03) | 0.64 SD (0.09) | 0.20 SD (0.05) | ✗ |
| | 40 | 0.78 (SD 0.04) | 0.81 (SD 0.08) | 0.25 (SD 0.03) | ✓ | 0.77 SD (0.02) | 0.80 SD (0.06) | 0.26 SD (0.06) | ✓ |
| | 50 | 0.76 (SD 0.04) | 0.78 (SD 0.09) | 0.25 (SD 0.04) | ✓ | 0.80 SD (0.04) | 0.87 SD (0.08) | 0.26 SD (0.05) | ✓ |
| | 60 | 0.79 (SD 0.02) | 0.82 (SD 0.04) | 0.23 (SD 0.04) | ✓ | 0.80 SD (0.03) | 0.84 SD (0.03) | 0.24 SD (0.05) | ✓ |
| | 70 | 0.73 (SD 0.08) | 0.69 (SD 0.19) | 0.23 (SD 0.07) | ✓ | 0.81 SD (0.06) | 0.86 SD (0.12) | 0.23 SD (0.05) | ✓ |
| | 80 | 0.77 (SD 0.02) | 0.81 (SD 0.04) | 0.26 (SD 0.03) | ✓ | 0.81 SD (0.01) | 0.85 SD (0.06) | 0.23 SD (0.05) | ✓ |
| | 90 | 0.78 (SD 0.03) | 0.79 (SD 0.07) | 0.23 (SD 0.02) | ✗ | 0.78 SD (0.02) | 0.78 SD (0.04) | 0.22 SD (0.04) | ✓ |
| Race | Asian | 0.79 (SD 0.08) | 0.83 (SD 0.12) | 0.24 (SD 0.11) | ✓ | 0.80 SD (0.11) | 0.84 SD (0.18) | 0.24 SD (0.08) | ✓ |
| | Black | 0.78 (SD 0.04) | 0.83 (SD 0.07) | 0.27 (SD 0.05) | ✓ | 0.80 SD (0.04) | 0.85 SD (0.07) | 0.24 SD (0.06) | ✓ |
| | Hispanic | 0.80 (SD 0.07) | 0.85 (SD 0.12) | 0.25 (SD 0.08) | ✓ | 0.80 SD (0.08) | 0.84 SD (0.16) | 0.25 SD (0.08) | ✓ |
| | Native | 0.78 (SD 0.17) | 0.97 (SD 0.07) | 0.43 (SD 0.35) | ✗ | 0.82 SD (0.13) | 1.00 SD (0.00) | 0.35 SD (0.23) | ✓ |
| | Other | 0.76 (SD 0.06) | 0.72 (SD 0.10) | 0.19 (SD 0.05) | ✓ | 0.79 SD (0.07) | 0.77 SD (0.09) | 0.20 SD (0.09) | ✓ |
| | Unknown | 0.79 (SD 0.05) | 0.83 (SD 0.11) | 0.25 (SD 0.03) | ✓ | 0.82 SD (0.03) | 0.87 SD (0.06) | 0.23 SD (0.05) | ✓ |
| | White | 0.77 (SD 0.02) | 0.79 (SD 0.06) | 0.24 (SD 0.03) | ✓ | 0.80 SD (0.02) | 0.84 SD (0.04) | 0.24 SD (0.05) | ✓ |
| Insurance | Medicaid | 0.72 (SD 0.07) | 0.69 (SD 0.17) | 0.26 (SD 0.06) | ✗ | 0.76 SD (0.08) | 0.77 SD (0.16) | 0.26 SD (0.05) | ✓ |
| | Medicare | 0.78 (SD 0.03) | 0.81 (SD 0.06) | 0.25 (SD 0.02) | ✓ | 0.81 SD (0.02) | 0.85 SD (0.04) | 0.24 SD (0.05) | ✓ |
| | Other | 0.78 (SD 0.02) | 0.80 (SD 0.05) | 0.24 (SD 0.03) | ✓ | 0.80 SD (0.02) | 0.84 SD (0.05) | 0.23 SD (0.04) | ✓ |
| Language | English | 0.77 (SD 0.04) | 0.79 (SD 0.09) | 0.25 (SD 0.04) | ✓ | 0.81 SD (0.06) | 0.85 SD (0.11) | 0.24 SD (0.05) | ✓ |
| | Other | 0.78 (SD 0.02) | 0.80 (SD 0.05) | 0.25 (SD 0.02) | ✓ | 0.77 SD (0.01) | 0.78 SD (0.04) | 0.24 SD (0.04) | ✓ |
| Marital status | Divorced | 0.78 (SD 0.04) | 0.80 (SD 0.10) | 0.24 (SD 0.03) | ✓ | 0.79 SD (0.05) | 0.82 SD (0.09) | 0.24 SD (0.05) | ✓ |
| | Married | 0.77 (SD 0.03) | 0.77 (SD 0.06) | 0.22 (SD 0.02) | ✓ | 0.81 SD (0.01) | 0.83 SD (0.05) | 0.22 SD (0.05) | ✓ |
| | Single | 0.78 (SD 0.02) | 0.84 (SD 0.05) | 0.28 (SD 0.03) | ✓ | 0.79 SD (0.03) | 0.84 SD (0.06) | 0.27 SD (0.04) | ✓ |
| | Widowed | 0.79 (SD 0.04) | 0.82 (SD 0.08) | 0.24 (SD 0.02) | ✓ | 0.81 SD (0.03) | 0.85 SD (0.07) | 0.24 SD (0.06) | ✓ |
| | Unknown | 0.77 (SD 0.05) | 0.83 (SD 0.09) | 0.29 (SD 0.05) | ✓ | 0.84 SD (0.04) | 0.93 SD (0.06) | 0.24 SD (0.07) | ✓ |

*SD* Standard deviation, *AUROC* Area under the receiver operating characteristic, *TPR* True positive rate, *FPR* False positive rate, *EO* Equalised odds

example, the TPR parity constraint on the short model achieved EO for those in the age bracket of 90. However, it caused the FPR of those in the minority group around 20 years old to increase from 0.29 to 0.61 (Supplementary Table. 5). This loss in performance for the 20-year-old group was also partially seen for the FPR parity constraint on the long model (Supplementary Table. 6). This shows the importance of balance when considering if a model is defined as fair or not, in particular for drastically different patient populations, such as 90 versus 20-year-olds. Prioritising one group or sensitive attribute can hinder model performance in others. As such honest and decent precautions and analysis are needed to ensure algorithms are equal and reasonable without discrimination. Moreover, for antibiotic decision making further ethical considerations need to be taken into account including the effect on other individuals outside of the patient being treated[33]. We believe this analysis demonstrates such CDSSs can be fair; however, further validation is certainly required.

Two feature sets were used in this research to evaluate the trade-off between simplicity and explainability vs performance which has been widely discussed in the machine learning literature[34]. Overall results show that the long model often demonstrates slightly superior performance to the short model. However, it is inherently more complex and in some scenarios such as in those with sepsis, it performs worse than the short model. Further research including understanding clinicians opinions is required to determine what model is most appropriate in specific circumstances. Alternative cutoff thresholds were also investigated for our binary classification task to maximise the AUROC and minimise the FPR. Results show that this was achieved for both the short and long models by fixing the thresholds from the preprocessing validation subset. With the 1st threshold achieving a reasonable AUROC and the 2nd threshold having a lower FPR, although as expected this comes at the expense of a worse AUROC score. We envisage such thresholds being utilised similar to a traffic light, whereby suggestions can be split into don't, potentially, or do switch based on the model's level of confidence (Fig. 4). This type of structure is simple, familiar to individuals and should ensure along with interpretability methods that such a model acts as an appropriate CDSS and allows for the end user to understand the output alongside other information in order to make the final decision.

Explainability and interpretability are critical aspects of using machine learning models in the real world[35,36]. To ensure our model and its outputs could be understood and interrogated SimplEx[29] was utilised and visual representations created (Fig. 4). These visual summaries include a number of aspects that were noted as important for understanding by clinical colleagues. Firstly textual descriptions enable key information to be conveyed quickly and reduce the barrier to adoption through universal understanding. Secondly, related patient examples are shown and scored. Clinicians rely heavily on prior experience when undertaking antibiotic treatment decisions[37]; as such, showing historical examples and how they compare to the current patient of interest is perceived as appropriate. In conjunction, highlighting whether the model was correct on previous examples at each threshold provides some level of reassurance on how well the model

performs on this type of patient and therefore if the predictions should be trusted or not. Finally, patient-specific feature contribution can be shown to illustrate how the model arrived at that conclusion. Figure 4 shows that while in many cases a clear switch decision is apparent, inherently some days (e.g., day 3) and patients present a particularly complex case. This reflects what is often seen in reality with decisions regarding antimicrobial switching not being clear-cut. By incorporating interpretability methods, models such as those developed in this research can become clinically useful CDSSs.

The objective of a CDSS to support IV-to-oral switch decision making is to facilitate antimicrobial stewardship. ASI results are in-line with current literature indicating frequent oral prescribing may use less broad spectrum IV antibiotics overall and therefore could be beneficial from an AMR and HCAIs perspective[12,38]. As such, this evidence supports the drive to maximise the use of oral therapies and alongside limited adoption[15,16] highlights why a switch focused CDSS may be useful. It is however notoriously difficult to discern the value of predictions from a CDSS. A retrospective analysis was conducted to understand how such switch models may benefit healthcare institutions and patients. Figure 2 shows for the first two days upon starting IV treatment our models predict that the majority of patients should not switch which corresponds with the true labels. This is in line with the latest UK guidelines whereby the IV-to-oral switch should be considered daily after 48 hours[14]. For dates with 2 to 7 prior days of IV treatment though there develops a disconnect between the labels and model predictions. This is particularly apparent for the short model and the first more lenient threshold. Model outputs indicate that by day 4 almost all patients could be suitable for switching to oral administration from a clinical parameter, health status perspective. For some patients, there will be risk factors beyond the models input features that the clinician considered meaning they did not switch, but for others, the clinician may have been unaware or neglected the decision meaning switching earlier may have been suitable. Furthermore, results show that LOS is minimised when predictions and the true labels align, and upon switching patients usually see prompt discharge. Our models may therefore be able to provide useful decision support by raising awareness of when switching could be suitable for a particular patient. Given this decision is often neglected and postponed, such a CDSS may be able to promote switching when appropriate which could potentially support efforts to stop AMR, prevent HCAIs, and benefit patients.

To improve the clinical applicability of our solution a number of logic-based rules could be implemented. For example, if a patient has a certain type of infection, malabsorption, immunosuppression, has recently vomited, or could have compliant issues, an overriding rule based on the latest guidelines[14] could suggest not to switch. Furthermore, the number of days of IV treatment should be highlighted alongside conditions, such as sepsis, in which extra care should be taken, as these factors influence switch decision making. If a patient is receiving an IV antibiotic and there is a similar oral version available this could be flagged alongside model outputs as a 'simple' switch. Moreover, given the potential comfort, workload, and discharge benefits when patients have no IV catheters, CDSSs should consider the wider patient treatment paradigm, and potentially further encourage switching when IV access is only for antibiotic treatment. Finally, to improve practice it is important for clinicians to document when a switch occurred and why that decision was made. This ensures in the future such individualised antibiotic decision making can be data-driven based on real evidence, rather than decided by habit or general population evidence. By combining machine learning approaches with clinical logic we can ensure patient safety while driving a positive change in antimicrobial utilisation. In the future we will conduct further research on how such solutions could be combined and implemented in real-time to create a complete CDSS for antibiotic optimisation, that is well received by the clinical community and provides novel, useful information.

There are limitations to this research study. Firstly, the use of historical patient data means that all of our models predictions are based on historical prescribing practices. Due to concerns surrounding AMR, there has been a large amount of research into antibiotic prescribing over recent years[17,39–42] and hence it is plausible our models switch suggestions are 'out of date'. Secondly, our model only analyses a snapshot of the patient and not all the factors that are clinically used to assess a patient's suitability for switching[14]. As discussed in the methods, this is due to data challenges, but incorporating additional criteria into the model so that under certain circumstances a switch suggestion cannot be given is an avenue for future work. However, we believe by anayzing and summarising multiple variables regarding the patients clinical and infection status such a system could support switch decision making with the final decision always made by the clinician. Finally, the current work presented only evaluates such models on US based ICU data. How such a system could perform in other medical settings and health-systems such as infectious diseases wards, the UK's NHS and low and middle income countries remains an outstanding question. But given the results presented and the routine, standardised nature of the raw input data we believe our approach is generalisable and there is potential to translate this research into other non-ICU medical settings where oral therapy may be more commonly utilised.

In summary, we have identified clinically relevant features and developed simple, fair, interpretable, and generalisable models to estimate when a patient could switch from IV-to-oral antibiotic treatment. In the future, this research will require further analysis and prospective evaluation to understand its safety, clinical benefit, and how it can influence antimicrobial decision making. But given AMR, HCAIs, and the interest in promoting oral therapies, such a system holds great promise to provide clinically useful antimicrobial decision support.

## Methods
### Datasets
Two publicly available large de-identified real-world clinical datasets containing routinely collected EHR information were used within this research. MIMIC-IV (4th version of the Medical Information Mart for Intensive Care database) which contains over 40,000 patients admitted to the Beth Israel Deaconess Medical Center (BIDMC) in Boston, Massachusetts between 2008 and 2019[21,22], was used for feature selection, model optimisation and hold out testing. Meanwhile, the eICU Collaborative Research Database contains data for over 200,000 admissions to ICUs across the United States from 2014 to 2015[22–24], was used for transfer learning to confirm genererlisability. Our study complies with all the data use and ethical regulations required to access the datasets. For both datasets, the patient population was filtered to those who received IV and oral antibiotic treatment within the ICU (IV treatment was limited to less than 8 days). Unfortunately, the datasets used in this research do not contain explicit information on if, when, or why an IV-to-oral switch was considered. However, by utilising the available prescribing data and taking what the clinicians actually did as a label we can approximate the prescribing behaviour and train a machine learning model. We therefore focused on making a route of administration prediction for each day the patient was on antibiotics given clinical decisions regarding antimicrobial treatment are most often made on a daily basis. As such negative switch labels were defined as each day a patient was on IV antibiotics, while positive labels were defined as every other day (i.e., where the patient was on oral but not IV antibiotics). The antibiotic spectrum index (ASI) from[25] was used to assess the average breadth of activity of IV and oral treatment regimes. By looking at the ASI on the day before switching and the first

day of only oral administration we can understand how a change in route of administration is most often associated with the ASI.

## Feature selection

Our aim was to make a model that through utilising routinely available patient vitals could act as a starting point for the decision making process and flag when a switch could be considered for a particular patient. The latest UK Health Security Agency (UKHSA) IVOS criteria[14] were analysed and ten related features were extracted from the datasets. Specifically: heart rate, respiratory rate, temperature, O2 saturation pulseoxymetry, systolic blood pressure, diastolic blood pressure, mean blood pressure, GCS motor response, GCS verbal response, and GCS motor eye opening (Supplementary Table 1). White Cell Count and C-Reactive Protein were excluded due to data missingness, requirement for a blood test, and UKHSA guidelines stating that they should be considered but are not necessary for a switch. Other important aspects of the guidelines such as infection type and absorption status, were also not included as input features to the model as much of this data was unavailable or collected in a way that makes it unsuitable for machine learning. Furthermore, evidence surrounding these is constantly changing[8,11,43]. We aimed to create a simple, generalisable model that uses only routinely available patient data and has the potential to be used in many different healthcare settings. The Canonical Time-series Characteristics (Catch22) methodology[44] (along with the mean and variance) was utilised through sktime[45,46] to transform temporal data into daily tabular values. This was done for each specific day and the whole of the current stay. In addition, the difference between transformed values for a given day and the preceding day was calculated. SHapley Additive exPlanations (SHAP) values[26] and a genetic algorithm[47] were then used for feature selection. Specifically, an excessively large neural network with 851,729 trainable parameters was preliminary trained, SHAP values were calculated and those features with a value of greater than or equal to 0.5 were selected for use in the genetic algorithm. The genetic algorithm optimised for AUROC and was run twice. Once for a simple set of 5 features and the second without a limitation on the number of features. 10 iterations with 50 individuals and 25 iterations with 20 individuals were used respectively.

## Model development

The MIMIC-IV EHR dataset was randomly split (50%, 50%) based on patients ICU 'stay_id' into a preprocessing and a hold-out set in order to generalise switching prescribing behavior and get a reliable unbiased estimate of the models performance given the selected hyperparameters and feature set. The preprocessing set was split randomly into training, validation, and testing sets for feature selection as discussed above with Pytorch[48] used to create the neural networks. After feature selection, optuna[49] with the objective of maximising the AUROC was used to select the models hyperparameters, and optimal alternative cutoff thresholds were determined from the preprocessing validation subset. Youden's Index[27] was used to optimise the AUROC, while finding the point where precision, recall and the F1 score were equal was used as a stringent cutoff for reducing the FPR. Subsequently, once the features and models were finalised the unseen hold-out set was randomly split 10 times into stratified training, validation, and testing sets for evaluation. Specifically, 10 naive models based on the previously identified features and model hyperparameters were trained and the final performance of such models was evaluated. The synthetic minority oversampling technique[50] was used during training to address label class imbalance. The Adam optimiser[51] was used with binary cross entropy with logits loss. The training utilised 10 epochs, and the model with the greatest AUROC on the validation dataset was selected as the final model to obtain results on the unseen test set.

## Model evaluation

Standard ML metrics were used to evaluate model performance. Specifically for the switch classification task the AUROC, accuracy, precision, TPR, FPR, F1 score, and Area Under the Precision Recall curve (AUPRC) were calculated. The standard deviation was calculated to indicate the variation in results. To provide a baseline for comparison two infection markers that are clearly defined within the latest guidelines[14] were separately also used for predicting when switching could be appropriate in each patient. Specifically, their temperature must have been between 36 °C and 38 °C for the past 24 h and the Early Warning Score must be decreasing, upon which a switch would then be suggested for the rest of that patient's stay. It was not possible to include every aspect of the guidelines due to many being ambiguous and not recorded within the data. However, it acts as a fair comparison to our models as it utilises similar patient data and actually contains additional information not fed into our models such as the inspired O2 fraction. The best performing final model and its respective hold-out split were used to break the distribution of labels and predictions down by IV treatment duration, to evaluate how predictions compare temporally to the real labels and discern when the model performed well vs poorly (Fig. 2). To understand the value of the models switch predictions and how they relate to patient outcomes, the difference in days between our predicted switch events and real switch events was calculated and the mean LOS and mortality outcomes were taken (Fig. 3, Supplementary Table 4). Furthermore, we analysed whether there was a variation in the remaining ICU length of stay (LOS) for patients who remained on IV vs those who switched on that day (Supplementary Fig. 3). This was done for dates with 2 to 7 days of previous IV treatment based on the dissimilarity between model predictions and labels on those days (Fig. 2). For statistical analysis the non-parametric Wilcoxon rank-sum (Mann-Whitney U) test with alpha set at 0.05 was used to test if the difference in means was statistical significant given the non-normal data distribution. Effect sizes were calculated using Cohen's d method with pooled standard deviation. Models were evaluated using functions and metrics from the Scikit-learn and SciPy libraries[52,53].

To further validate findings, evaluations were performed on specific patient populations and infectious diseases within MIMIC. Antibiotics with incomplete oral absorption (bioavailability < 90%) were determined through consultation with a pharmacist and a literature search on PubMed, the Electronic Medicines Compendium, and UpToDate. The final list of antibiotics with incomplete oral absorption is shown in Supplementary Table 7. Total parenteral nutrition was used as a proxy for poor oral absorption (malabsorption) while hospital ICD diagnostic codes were used to identify patients with UTI's, pneumonia, and sepsis. These infections were chosen as they are highly prevalent in the dataset and UTI's/pneumonia are commonly treated with oral antibiotics but sepsis sees less oral utilisation. Note that infection types in MIMIC are linked to the hospital stay 'hadm_id' only and not the specific ICU stay 'stay_id' as diagnoses are only coded for billing purposes upon hospital discharge. The best performing short and long models from the MIMIC hold-out set were then evaluated on data extracted from the eICU database via transfer learning to re-train and subsequently test the models. The same data processing pipeline was used for eICU and transfer learning utilised the same procedure as with evaluation on the MIMIC hold-out set except the models parameters were initialised with the best performing final MIMIC trained models.

The best performing final short and long models trained on the MIMIC hold-out set were used for fairness and interpretability research. SimplEx[29] was used as a post-hoc explanation methodology to extract similar patient examples, their importance, and the contribution of each feature for each example. To this extent first, the corpus and test latent representations are computed. SimplEx was

then fitted and the integrated jacobian decomposition for a particular patient was calculated and displayed. To simplify visualisations only those examples with an importance greater than 0.1 are shown. To assess model fairness the demanding equalised odds (EO) metric was used given we want to acknowledge and ideally minimise false positives as well as obtain equal performance across sensitive attribute classes. We defined that EO was achieved for a given sensitive attribute group if the TPR was not less than 0.1 from the global average and the FPR was not greater than 0.1 from the global average. EO was assessed utilising the 1st threshold for the sensitive attributes age (grouped into brackets based on the nearest decade), sex, race, insurance, language, and marital status. Threshold optimisation[30] was then employed to see if the models fairness could be improved. Specifically, the post-processing thresholdoptimizer method from fairlearn[54] was used with the balanced accuracy objective and either the equalised odds, FPR parity or TPR parity constraint.

### Reporting summary

Further information on research design is available in the Nature Portfolio Reporting Summary linked to this article.

## Data availability

Publicly available datasets were analyzed in this study. The MIMIC-IV dataset can be found at https://physionet.org/content/mimiciv/2.0/ and the eICU dataset at https://physionet.org/content/eicu-crd/2.0/. Both are accessible once you are a credentialed user on physionet, have completed the required training and signed the appropriate data use agreement. Specific additional data can be provided upon request to the authors, provided that it is in line with the datasets data use and ethical regulations. Source data are provided with this paper.

## Code availability

The computer code used in this research is available at https://github.com/WilliamBolton/iv_to_oral[55].

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

## Acknowledgements

William Bolton was supported by the UKRI CDT in AI for Healthcare http://ai4health.io (Grant No. P/S023283/1) and by the National Institute for Health, Health Protection Research Unit in Healthcare Associated Infections and Antimicrobial Resistance at Imperial College London in partnership with the UK Health Security Agency (previously PHE), in collaboration with, Imperial Healthcare Partners, University of Cambridge and University of Warwick. He is also affiliated to the Department of Health and Social Care, Centre for Antimicrobial Optimisation. The authors would like to acknowledge (1) the National Institute for Health Research Health Protection Research Unit (NIHR HPRU) in Healthcare Associated Infection and Antimicrobial Resistance at Imperial College London and (2) The Department for Health and Social Care funded Centre for Antimicrobial Optimisation (CAMO) at Imperial College London. This study is independent research partly funded by the National Institute for Health Research. The views expressed in this publication are those of the authors and not necessarily those of the NHS, the National Institute for Health Research the Department of Health and Social Care or the UK Health Security Agency.

## Author contributions

W.B., R.W., and T.M.R. contributed to study concept and design. W.B. contributed to data acquisition and analysis. W.B. and T.M.R. contributed to the manuscript drafting, discussion of the results, and review of the data. All authors contributed to data interpretation, as well as final revisions of the manuscript. All authors had full access to all the data in the study and had final responsibility for the decision to submit for publication.

## Competing interests

Author T.M.R. was employed by Sandoz (2020), Roche Diagnostics Ltd (2021), and bioMerieux (2021–2022). These commercial entities were not involved in the study design, collection, analysis, interpretation of data, the writing of this article or the decision to submit it for publication. All authors declare no other competing interests.
