## [Peer Review File · Nature Communications]

Reviewers' Comments:

Reviewer #1:

Remarks to the Author:

In this manuscript, the authors tackle an important problem of encouraging clinicians to transition from IV to oral antibiotic therapy using an ML-based CDSS. The strengths of the paper are that they take care to address interpretability, assess fairness, and choose their feature sets in an unbiased manner with a genetic algorithm. The main weaknesses are an apparent lack of generalizability and a lack of explanation for why this is. I also struggle to see how a model trained on and designed to mimic real-world clinical practice can really promote early switching from IV to oral antibiotics. The authors claim that the model provides personalized recommendations, but its features are basic vital signs and it doesn't seem to consider more complicated patient risk factors that cannot be easily represented numerically.

1. How does the ML model performance compare to a baseline like using standard clinical guidelines for switching? An example of those guidelines could be found in Carratala et al., JAMA Internal Medicine, 2012. "Patients were switched from IV to oral therapy when they experienced clinical improvement and met the following objective criteria: ability to maintain oral intake; stable vital signs (considered as temperature $\leq 37.8^{\circ}\text{C}$, respiratory rate ≤ 24 breaths/min, systolic blood pressure ≥ 90 mm Hg without vasopressor support for at least 8 hours); and absence of exacerbated major comorbidities (ie, heart failure, chronic obstructive pulmonary disease) and/or septic metastases."

2. The stated goal of the authors is to promote antimicrobial stewardship - a worthy goal indeed! However, the results from the eICU hold-out dataset call into question whether the model can really achieve that goal, in that the model predicts that far fewer patients should be switching to oral therapy. Do the authors agree with that interpretation, and can they explain why that might be? This finding surprises me because there seem to be more patient characteristics (co-morbidities, administration of vasopressor support that artificially increases the blood pressure) that are not considered by the model that might cause a clinician to prolong IV therapy than vice versa. What are the differences between the MIMIC and eICU patient characteristics or clinical practices that might lead to that discrepancy?

3. This is a bold claim that is not fully supported: "We can extrapolate that our models may be able to promote early switching when appropriate and that this potentially could support efforts to stop AMR, prevent HCAs and benefit patients." The model is trained on current medical practice, how can it promote "early" switching?

4. What is the clinical cause of discrepancies between the model prediction and real life in the held-out test sets? That is, when the clinicians maintain IV therapy but the model recommends a switch, are there certain patient risk factors that the clinicians considered that the model is blind to? And vice versa, did the clinicians just neglect to make the switch to oral? Other ML-based CDSS papers have approached this question by performing clinical manual review of ~ 20 randomly selected patient charts or applying their model prospectively in a small cohort.

5. Regarding Fig. 4, I applaud the authors' efforts to provide interpretability but the feature values (beyond "SpO2") are opaque. Can the authors more clearly explain the top 5 features in the context of their Catch22 time-series transformation? For example, what's the difference between feature 4 (O2 sat pulseox with a catch22 transformation of 3 for the entire stay) and one of the additional 32 features (O2 sat pulseox with a catch 22 transformation of 1 for the entire stay)? I also find the reporting of two thresholds to be confusing, can the authors clarify the benefit of using both thresholds?

6. Please clarify the role of Table 6. The drugs listed (e.g. amoxicillin, Augmentin, azithromycin, ciprofloxain, etc.) are routinely given orally without issue, and the biggest influencer of antibiotic choice is the organism that the patient is infected with.

7. Since the MIMIC-IV dataset contains data from 2008-2019, how does the model perform if it's trained on older data and tested on newer data? Splitting the data this way would be more

representative of how the model would perform if it were to be implemented in the real world.

8. Could the authors provide a table on the patient demographics in the 3 datasets (MIMIC preprocessing, MIMIC hold-out, and eICU hold-out)? e.g. #, age, race, clinical syndrome (e.g. sepsis), organism, LOS, days on IV antibiotics, days on oral antibiotics, IV therapy regimen, oral therapy regimen, etc.?

Reviewer #2:

Remarks to the Author:

Overall and interesting approach to a process that is complex and that we train for many years to learn the skills, judgement and expertise to perform. As such my comments come with the consideration that although this CDSS can add some perspective to the process, though it falls short of considering other crucial aspects that are needed (see detailed comments below).

Further the context in which it is used needs careful consideration. If we provide junior staff, inexperienced clinicians with a decision tool that provides a traffic light output, without the necessary information that is needed to be added to this (clinical condition, comorbidities of patients, immunosuppression) there is a risk it will be followed regardless, sometimes in error.

Abstract

Some clinical parameters are assessed here, but not all that we would use to assess suitability to switch. Observations you have used are used as a marker of overall improvement, but as the UKHSA document suggests so do WCC and CRP.

Points from Intro and Discussion

The heterogeneity of patient infections, comorbidities and immunosuppression are not discussed - these are crucial factors that will impact our decisions of when to switch

You have not included assessing gut absorption, essential if oral medications will work

It would be helpful in the introduction to outline why IV abx are used at the outset - bioavailability, sepsis, central nervous system infection. Also please note that oral is non-inferior for SOME infections, but not all (endocarditis, sepsis, central nervous system infection, bone and joint)

Results

Patient cohort - need to explain who these patients were (adults) and where they were located (US) as context matters. No information is given about the types of infections these patients had, and if there was an assessment as to IF they were suitable for oral switch (see above point). Or if excluded, which ones were excluded?

Criteria selection - 10 features stated but cannot see where this is from. There is a UKHSA flow chart with a number of points to assess and they have only used items from points 3.1 and 3.2. Need to communicate in the manuscript very clearly that clinical observations only were considered, it is not highlighted enough that this is only a snapshot of the patient, not a full assessment.

figure 1 -put all the acronyms here so not needing to look up in the main text, they are not commonly used

figure 4 - output is a nice visualisation, attractive and clear, but need to state very clearly what is not being considered (specific infection type, comorbidities, immunosuppression, gut function)

Methods

These need to be sufficiently detailed to explain how you have developed your algorithm, but as it stands, non experts are unlikely to understand what you have done and therefore have confidence

to use this. It may need further dissemination plans if this is the case at a later stage.

Thank you for your review, your comments and feedback are very much appreciated. We have implemented the requested changes into the manuscript and noted our response and action to your comments below. Please note that page numbers refer to the manuscript with the changes highlighted.

Reviewer #1

Comment: In this manuscript, the authors tackle an important problem of encouraging clinicians to transition from IV to oral antibiotic therapy using an ML-based CDSS. The strengths of the paper are that they take care to address interpretability, assess fairness, and choose their feature sets in an unbiased manner with a genetic algorithm. The main weaknesses are an apparent lack of generalizability and a lack of explanation for why this is. I also struggle to see how a model trained on and designed to mimic real-world clinical practice can really promote early switching from IV to oral antibiotics. The authors claim that the model provides personalized recommendations, but its features are basic vital signs and it doesn't seem to consider more complicated patient risk factors that cannot be easily represented numerically.

Response: Thank you for your comment. We approached this problem from the perspective of utilising routinely available data. Our model is still able to provide personalised recommendations based on a patient's historic and individual vital signs at a given point in time. We appreciate a limitation is that our model does not cover everything required for an IV to oral switch decision, this is because much of this data is not available or collected in a way that makes it unsuitable for machine learning. However, we believe our model could be useful as a starting point for switch decision making to highlight when a switch could be considered for a particular patient. Regarding generalizability and early switching we have commented and addressed these points below.

Action: Clarified the text to note the limitations of our model (page 14).

Comment: 1. How does the ML model performance compare to a baseline like using standard clinical guidelines for switching? An example of those guidelines could be found in Carratala et al., JAMA Internal Medicine, 2012. "Patients were switched from IV to oral therapy when they experienced clinical improvement and met the following objective criteria: ability to maintain oral intake; stable vital signs (considered as temperature $\leq 37.8^{\circ}\text{C}$, respiratory rate ≤ 24 breaths/min, systolic blood pressure ≥ 90 mm Hg without vasopressor support for at least 8 hours); and absence of exacerbated major comorbidities (ie, heart failure, chronic obstructive pulmonary disease) and/or septic metastases."

Response: We have added a comparison to the current UK guidelines for those variables that were available and can be transformed into a clear prediction. Specifically using the rules: 'Temperature has been between 36 to 38°C for the past 24 hours' and 'Early Warning Score is decreasing'. It is not possible to compare all aspects of the guidelines as many are ambiguous, not objective and are infrequently recorded and hence are intractable to assess from a database / machine learning perspective. We did not use the guidelines from Carratala et al. as they study a very different patient population (Community acquired pneumonia vs ICU).

Action: Added baseline comparison to current guidelines (pages 5, 6, and 16).

Comment: 2. The stated goal of the authors is to promote antimicrobial stewardship - a worthy goal indeed! However, the results from the eICU hold-out dataset call into question whether the model can really achieve that goal, in that the model predicts that far fewer patients should be switching to oral therapy. Do the authors agree with that interpretation, and can they explain why that might be? This finding surprises me because there seem to be more patient characteristics (co-morbidities, administration of vasopressor support that artificially increases the blood pressure) that are not considered by the model that might cause a clinician to prolong IV therapy than vice versa. What are the differences between the MIMIC and eICU patient characteristics or clinical practices that might lead to that discrepancy?

Response: We agree that in the eICU dataset the model predicts fewer patients should switch than in MIMIC. There are several potential reasons for this. Firstly, the patients in eICU are on average more severely unwell than MIMIC as demonstrated in the demographics table (page 4), therefore understandably the model should predict less are suitable for oral administration. Secondly eICU is a USA wide dataset containing many different hospitals and hence prescribing behaviour with regards to IV to oral switching may be more heterogeneous than in MIMIC whose data is from a single institution. Switching administration route is influenced by many behavioural factors. As such, the eICU model is having to approximate many different behaviours into one, which likely results in it being more conservative with regards to predicting switching to optimise performance.

With regards to generalisability we believe that the model is generalizable as it is able to obtain similar performance across different patient populations and through utilising the same data processing pipeline across two distinct datasets (MIMIC and eICU). As discussed above though the switching behaviour of the institution is very important as such before implementation, we envisage the model being tuned for each hospital. A practice that is very common in machine learning applications. Furthermore, by focusing the research on routinely collected data in the form of major patient vitals, we are inherently making the model generalizable as it has the potential to be used in many settings. More complex data on gut function, intubation status and infection type are often not digitally recorded and so could limit the generalizability of a model.

Action: Added demographics table (page 4). Clarified the rationale behind eICU predictions (page 11). Clarified why our model is generalizable (pages 11 and 12).

Comment: 3. This is a bold claim that is not fully supported: "We can extrapolate that our models may be able to promote early switching when appropriate and that this potentially could support efforts to stop AMR, prevent HCAs and benefit patients." The model is trained on current medical practice, how can it promote "early" switching?

Response: Unfortunately, the datasets used in this research do not have information on if, when or why an IV to oral switch was considered. As such, we must make assumptions based on the prescribing data available. A range of behavioral factors influence the decision to

switch and are not easily accounted for in the data. However, by taking what the clinicians actually did as a label we can approximate these and model when switching occurred. Even though the model is trained on historical data it could potentially still promote early switching by raising awareness on when switching could be suitable, as this decision is often neglected and postponed given the current guidelines are based on little evidence and clinicians can often be hesitant. However, we appreciate this is a complex point and requires more evidence to be claimed and hence have removed the word 'early'.

Action: We have revised the wording throughout the manuscript to reflect this. We have clarified in the discussion (page 13) why our model could act as an appropriate decision support system for when to consider switching or not.

Comment: 4. What is the clinical cause of discrepancies between the model prediction and real life in the held-out test sets? That is, when the clinicians maintain IV therapy, but the model recommends a switch, are there certain patient risk factors that the clinicians considered that the model is blind to? And vice versa, did the clinicians just neglect to make the switch to oral? Other ML-based CDSS papers have approached this question by performing clinical manual review of ~20 randomly selected patient charts or applying their model prospectively in a small cohort.

Response: The current model does not take into account all the factors that influence a decision to switch. For example, type of infection, antibiotic and immunosuppression status. We appreciate this is a limitation, but these factors were not included due to missing data and because we aimed to create a generalizable tool upon which more rigid rule like criteria could be added in the future. However, this combined with non-optimal stewardship from clinicians, means both scenarios you mentioned probably occur. Given the large behavioural variation and lack of gold standard opinion on decision making in this space, manual review at this stage is unlikely to yield informative results. In the future we plan to undertake prospective testing of such as system to further understand this point, and how such a system would be perceived by clinicians and could potentially influence decision making.

Action: We have clarified this in the limitations section (page 14). We have clarified the generalizability of the model (pages 11 and 12). We have noted potential reasons for the discrepancies between model predictions and the labels (page 13). We have noted prospective testing as future work (page 14).

Comment: 5. Regarding Fig. 4, I applaud the authors' efforts to provide interpretability but the feature values (beyond "SpO2") are opaque. Can the authors more clearly explain the top 5 features in the context of their Catch22 time-series transformation? For example, what's the difference between feature 4 (O2 sat pulseox with a catch22 transformation of 3 for the entire stay) and one of the additional 32 features (O2 sat pulseox with a catch 22 transformation of 1 for the entire stay)? I also find the reporting of two thresholds to be confusing, can the authors clarify the benefit of using both thresholds?

Response: We have explained the top 5 features and their relevance for IV to oral decision making in the first paragraph of the discussion (page 11) and note the different types of transformations in table 5. For example, O2 sat pulseox with a catch22 transformation of 1

uses the DN_HistogramMode_10 transformation while O2 sat pulseox with a catch22 transformation of 3 uses CO_FirstMin_ac. Unfortunately given the nature of the catch22 time series transformations, the features themselves are not immediately intuitive. To address this, we have clarified in the text the key and where to find further information on the different catch22 transformations. Regarding interpretability to account for the non-intuitive features, we display similar patients so that their values can be compared and envisage that in reality the raw data (e.g., current and historical O2 sat) would be presented in a scatter graph / table alongside the model so the clinicians can view everything together.

Two thresholds were used to maximise the AUROC and minimise the FPR. This then allowed for a traffic light system to be used for simplicity and interpretability.

Action: We have clarified the catch22 transformation key and where to find more information (pages 24 and 25). We have specified why two thresholds were used (pages 5, 8, and 16).

Comment: 6. Please clarify the role of Table 6. The drugs listed (e.g., amoxicillin, augmentin, azithromycin, ciprofloxain, etc.) are routinely given orally without issue, and the biggest influencer of antibiotic choice is the organism that the patient is infected with.

Response: The aim of this experiment was to see if the model still performed well for those patients who were on oral drugs without complete absorption. As we wanted to ensure the model works in these cases where clinically you accept a degree of reduced exposure. This table was used to note drugs included in this test. We agree with you that these drugs are often used and that the biggest influence of antibiotic is the infective organism and its resistance profile.

Action: We have reframed the wording throughout from focusing on bioavailability to incomplete absorption (pages 6, 7, and 17).

Comment: 7. Since the MIMIC-IV dataset contains data from 2008-2019, how does the model perform if it's trained on older data and tested on newer data? Splitting the data this way would be more representative of how the model would perform if it were to be implemented in the real world.

Response: In this research data splitting was performed randomly based on patients 'stay_id' to generalise the switch prescribing behaviour. We appreciate the advantages of a temporal split and will investigate this in future work. In particular, through prospective testing via data collected at our local hospitals.

Action: Clarified data split was random (page 16). Noted prospective testing as future work (page 14).

Comment: 8. Could the authors provide a table on the patient demographics in the 3 datasets (MIMIC preprocessing, MIMIC hold-out, and eICU hold-out)? e.g., #, age, race, clinical syndrome (e.g. sepsis), organism, LOS, days on IV antibiotics, days on oral antibiotics, IV therapy regimen, oral therapy regimen, etc.?

Action: We have added a table on the demographics and most common antibiotics (of each dataset (page 4).

Reviewer #2

Comment: Overall and interesting approach to a process that is complex and that we train for many years to learn the skills, judgement, and expertise to perform. As such my comments come with the consideration that although this CDSS can add some perspective to the process, though it falls short of considering other crucial aspects that are needed (see detailed comments below).

Response: Thank you for your comment we appreciate your expert clinical viewpoint and the limitations of our research. We hope our responses and actions below overcome your concerns.

Comment: Further the context in which it is used needs careful consideration. If we provide junior staff, inexperienced clinicians with a decision tool that provides a traffic light output, without the necessary information that is needed to be added to this (clinical condition, comorbidities of patients, immunosuppression) there is a risk it will be followed regardless, sometimes in error.

Response: We approached this problem from the perspective of utilising routinely available data. We appreciate a limitation is that our model does not cover everything required for an IV to oral switch decision, this is because much of this data is not available or collected in a way that makes it unsuitable for machine learning. Other factors such as type of infection and immunosuppression should obviously be considered when making a switch decision. However, our aim was to make a model that could act as a starting point for the switch decision making process and flag when a switch could be considered such a tool would be generalizable and more rigid rule like criteria could be added in the future. Specifically, additional criteria can be added to the model so that under certain conditions a switch recommendation is not given for example if the patients are severely immunocompromised. The current limitation preventing this is data availability. We have discussed this in the discussion section (pages 13/14). As such, we are aligned that it would act as a decision support tool only and should certainly not be followed blindly. In addition, by adding interpretability methods we hope to ensure the models are not followed out of habit but instead are an informative part of the information considered when deciding whether to switch or not.

Action: Clarified the text to note the limitations of our model (page 14). Adjusted text to highlight potentially adding additional criteria in the future (pages 13/14). Clarified the decision support tool would act as the starting point of the switch decision making process (pages 12 and 15).

Comment: Abstract. Some clinical parameters are assessed here, but not all that we would use to assess suitability to switch. Observations you have used are used as a marker of overall improvement, but as the UKHSA document suggests so do WCC and CRP.

Response: As discussed above we focused the model on routinely available data and note that it does not cover everything required for an IV to oral switch decision. Regarding WCC and CRP as per the UKHSA guidelines these non-specific inflammation markers should be considered but are not required for switch. Given this we did not want the model to use these features as it could lead to an appropriate switch decision being missed or incorrect predictions if this data was not available. Furthermore, CRP was frequently missing in the dataset and so could not be used.

Action: We clarified in the text why the non-specific inflammation markers were excluded (page 15).

Comment: Points from Intro and Discussion The heterogeneity of patient infections, comorbidities and immunosuppression are not discussed - these are crucial factors that will impact our decisions of when to switch

Response: We appreciate the model does not cover every aspect of the switch decision making process and have noted this as a limitation. We wanted to approach this problem from a holistic perspective where a recommendation could be given regardless of antibiotic or type of infection similar to the current guidelines. Other factors in the decision-making process such as immunosuppression are very important but are ambiguous and not clearly defined from a historical ICU database perspective, meaning they are currently intractable to use.

Furthermore, in separate research we are actively working on understanding how comorbidities and antibiotic usage are related and how to best feed this information into AI decision support systems. But this is beyond the scope of this text.

Action: Clarified the text to note the limitations of our model (page 14). Added a point in the introduction to note the key variables for switch decision making (page 2). Noted the reasons behind our research approach and feature selection (page 15).

Comment: You have not included assessing gut absorption, essential if oral medications will work

Response: We tested our model on a group with total parenteral nutrition as a proxy for poor gut absorption. These results show that the model can still predict well in this cohort. Detailed gut absorption data is unfortunately not readily available in the dataset Furthermore, recent literature has indicated that some patients can still absorb oral antibiotics with poor gut absorption (

<https://academic.oup.com/jac/advance-article/doi/10.1093/jac/dkad198/7214002>).

Therefore, we wanted our model to be agnostic to this to allow for the clinician to incorporate this type of information into their final decision.

Action: We have added this reference to the text (page 15).

Comment: It would be helpful in the introduction to outline why IV abx are used at the outset - bioavailability, sepsis, central nervous system infection. Also please note that oral is non-

inferior for SOME infections, but not all (endocarditis, sepsis, central nervous system infection, bone and joint)

Response: We appreciate that oral antibiotics are not always suitable and have now added further information on why IV antibiotics are used at the outset in the introduction. However, for more and more types of infections it is being shown that orals can be suitable. Please see references the references from Platts et al, Kaasch et al, Spellberg et al, Iversen et al, Li et al, and Wald-Dickler et al. We therefore wanted to approach this problem from a holistic perspective where a recommendation could be given regardless of antibiotic or type of infection similar to the current guidelines. This allows for the clinician to decide how to incorporate this type of information into their final decision.

Action: We have added more information on why IV antibiotics are used at the outset (page 2). We have added more information on oral antibiotics and the new types of infection in which it can be non-inferior (page 2).

Comment: Results Patient cohort - need to explain who these patients were (adults) and where they were located (US) as context matters. No information is given about the types of infections these patients had, and if there was an assessment as to IF they were suitable for oral switch (see above point). Or if excluded, which ones were excluded?

Response: We have added more information on the datasets for further context on the cohort via a table. Unfortunately, we do not have any information on whether or not an IV to oral switch was considered at any given time point and so we must make assumptions based on the antibiotic prescribing data available.

Action: We have added a table on the demographics and most common antibiotics (of each dataset (page 4).

Comment: Criteria selection - 10 features stated but cannot see where this is from. There is a UKHSA flow chart with a number of points to assess and they have only used items from points 3.1 and 3.2. Need to communicate in the manuscript very clearly that clinical observations only were considered, it is not highlighted enough that this is only a snapshot of the patient, not a full assessment.

Response: We agree this model only assesses a snapshot of the patient. As discussed above we believe it could aid in decision making by analysing and summarising multiple clinical variables, but the final decision would always be with the clinician. Note that the model will also cover point 2.1 of the guidelines as the temporal data will contain information on if the patients clinical signs and symptoms are improving.

Action: Clarified the 10 features investigated in the methods section (page 15). Clarified the text to note the limitations of our model and that it only assesses a snapshot of the patient (page 14). Clarified the aim of the model was to only utilise routinely available patient vitals (page 15).

Comment: figure 1 -put all the acronyms here so not needing to look up in the main text, they are not commonly used

Action: Added acronyms to the figure and table legends.

Comment: figure 4 - output is a nice visualisation, attractive and clear, but need to state very clearly what is not being considered (specific infection type, comorbidities, immunosuppression, gut function)

Action: Clarified the text to note the limitations of our model (page 14). Adjusted the figure to note its limitations and use case (page 9).

Comment: Methods These need to be sufficiently detailed to explain how you have developed your algorithm, but as it stands, non-experts are unlikely to understand what you have done and therefore have confidence to use this. It may need further dissemination plans if this is the case at a later stage.

Response: Thank you for your comment we have reviewed and refined the methods section to provide sufficient detail. We agree that dissemination will be very important, and, in the future, we will undertake prospective testing to understand how to best present the model and its predictions in a clear while technical way to ensure appropriate utilisation.

Action: Reviewed and refined methods section to provide sufficient detail.

Reviewers' Comments:

Reviewer #1:

Remarks to the Author:

Thanks for your responses and changes. Please add how you calculated the baseline to the methods (I think the manuscript currently only refers to "two clear infection markers from the latest guidelines").

Reviewer #2:

Remarks to the Author:

I am happy with the edits made in response to my comments, which now make the manuscript better as they convey the uncertainty of the process and more information to caveat the system for interpretation of results.

Thank you for your review, your comments and feedback are very much appreciated. We have noted our response to your comments below.

Reviewer #1

Comment: Thanks for your responses and changes. Please add how you calculated the baseline to the methods (I think the manuscript currently only refers to "two clear infection markers from the latest guidelines").

Response: Thank you for your comment, we provide details on how the baseline was calculated in the first paragraph of the "Model evaluation" subsection in methods (page 11).

Action: Clarified the infection markers used in the statement "two clear infection markers from the latest guidelines" on page 4.

Reviewer #2

Comment: I am happy with the edits made in response to my comments, which now make the manuscript better as they convey the uncertainty of the process and more information to caveat the system for interpretation of results.

Response: Thank you for your comment we appreciate your review.